

# Long term dynamics of OH* temperatures over Middle Europe: Trends and solar correlations

Christoph Kalicinsky[1], Peter Knieling[1], Ralf Koppmann[1], Dirk Offermann[1], Wolfgang Steinbrecht[2], and Johannes Wintel[1]

[1]Institute for Atmospheric and Enviromental Research, University of Wuppertal, Germany
[2]DWD, Hohenpeissenberg Observatory, Germany

*Correspondence to:* C. Kalicinsky (kalicins@uni-wuppertal.de)

**Abstract.** We present the analysis of annual average OH* temperatures in the mesopause region derived from measurements of the GRound based Infrared P-branch Spectrometer (GRIPS) at Wuppertal ($51°$ N, $7°$ E) in the time interval 1988 to 2015. The current study uses a 7 year longer temperature time series compared to the latest analysis regarding the long term dynamics of OH* temperatures measured at Wuppertal. This additional time of observation leads to a change in characterisation of the observed long term dynamics.

We perform a multiple linear regression using the solar radio flux F10.7cm (11-year cycle of solar activity) and time to describe the temperature evolution. The analysis leads to a linear trend of $(-0.089 \pm 0.055)$ K year$^{-1}$ and a sensitivity to the solar activity of $(4.2 \pm 0.9)$ K $(100\,\mathrm{SFU})^{-1}$ ($r^2$ of fit 0.6). However, one linear trend in combination with the 11-year solar cycle is not sufficient to explain all observed long term dynamics. Actually we find a clear trend break in the temperature time series in middle of 2006. Before this break point there is an explicit negative linear trend of $(-0.22 \pm 0.08)$ K year$^{-1}$ and after 2006 the linear trend turns positive with a value of $(0.38 \pm 0.23)$ K year$^{-1}$. This apparent trend break can also be described using a long periodic oscillation. One possibility is to use the 22-year solar cycle that describes the reversal of the solar magnetic field (Hale cycle). A multiple linear regression using the solar radio flux and the solar polar magnetic field as parameters leads to the regression coefficients $C_{solar} = (5.0 \pm 0.7)$ K $(100\,\mathrm{SFU})^{-1}$ and $C_{hale} = (1.8 \pm 0.5)$ K $(100\,\mathrm{\mu T})^{-1}$ ($r^2 = 0.71$). But the best way to describe the OH* temperature time series is to use the solar radio flux and a 24-year oscillation. A multiple linear regression using these parameters leads to a sensitivity to the solar activity of $(4.3 \pm 0.7)$ K $(100\,\mathrm{SFU})^{-1}$ and an amplitude of the 24-year oscillation $A = (1.95 \pm 0.43)$ K ($r^2 = 0.77$). The most important finding here is that using these parameters for the multiple linear regression an additional linear trend is no longer needed. Moreover, with the knowledge of this 24-year oscillation the linear trends derived in this and in a former study of the Wuppertal data series can be reproduced by just fitting a line to the corresponding part (time interval) of the oscillation. This actually means that depending on the analysed time interval completely different linear trends with respect to magnitude and sign can be observed. This fact is of essential importance for any comparison between different observations and model simulations.

After detrending the temperature time series regarding the 11-year solar cycle and the 24-year oscillation multi-annual oscillations (MAOs) remain. A harmonic analysis finds three pronounced oscillations with periods of $(2.69 \pm 0.06)$ years,



$(3.15 \pm 0.07)$ years, and $(4.54 \pm 0.17)$ years. The corresponding amplitudes are $(1.03 \pm 0.33)$ K, $(1.03 \pm 0.33)$ K, and $(0.91 \pm 0.36)$ K, respectively.

## 1 Introduction

The mesopause of the Earth is one of the most variable regions in the atmosphere. There are numerous different influences such as the solar radiation and different types of waves (e.g. tides, planetary waves, gravity waves) that affect the temperature in this region. Thus, the temperature undergoes large variations on very different timescales from minutes to years. The largest variation observed in temperature is the variation during one year. This seasonal variation is characterised by an annual, a semi-annual, and a ter-annual component (see e.g. Bittner et al., 2000) and shows maximum to minimum temperature differences of up to 60 K throughout a year (see Fig. 1). The second largest temperature variations are caused by different types of waves. The induced temperature fluctuations occur on timescales from days up to months in case of planetary waves (e.g. Bittner et al., 2000; Offermann et al., 2009; Perminov et al., 2014) and on the timescale of several minutes in case of gravity waves (e.g. Offermann et al., 2011; Perminov et al., 2014). Beside these rather short term fluctuations the temperature in the mesopause region also exhibits long term variations on the timescale of several years. Although the amplitudes of these long term variations are much smaller, the long term change of the mesopause temperatures is nevertheless clearly existent and important. Several previous studies showed the existence of an 11-year modulation of the temperature in coincidence with the 11-year cycle of solar activity which is visible in the number of sunspots and the solar radio flux F10.7cm (for a review of solar influence on mesopause temperature see Beig , 2011a). The reported sensitivities in the mid- to high-latitudes of the northern hemisphere lie between 1 to $6 \, \mathrm{K} \, (100 \, \mathrm{SFU})^{-1}$. Another type of long term change are linear trends in the analysed time interval. In the mesopause region of the northern hemisphere such trends range between about zero trend up to a cooling of $3 \, \mathrm{K} \, \mathrm{decade}^{-1}$ (for a review of mesopause temperature trends see Beig , 2011b). Also trend breaks seems to be, where the linear trend switches its sign (positive or negative trend) or the magnitude of the trend significantly changes (for an example of the latter case see Offermann et al., 2010).

Beside these variations of the mesopause temperature Höppner and Bittner (2007) found a quasi 22-year modulation of the planetary wave activity which they derived from mesopause temperature measurements. This observed modulation coincides with the reversal of the solar polar magnetic field, the so-called Hale cycle. The solar polar magnetic field reverses every approximately 11 years at about solar maximum and, thus, the maximum positive and negative values of magnetic field strength occur in between two consecutive solar maxima (e.g. Svalgaard et al. , 2005). Several studies showing a quasi 22-year modulation of different meteorological parameters such as temperature, rain fall, and temperature variability that are in phase with the Hale cycle or the double sunspot cycle (another type of Hale cycle with a period of about 22 years which is phase shifted compared to the Hale cycle of the solar polar magnetic field; the maxima and minima of the double sunspot cycle occur at maxima of the sunspot number (e.g. King , 1975; Qu et al. , 2012)) exist (e.g. Willet , 1974; King et al. , 1974; King , 1975; Qu et al. , 2012), but no physical mechanism is found for these coincidences. However, a number of possible influences also showing a 22-year modulation are named: galactic cosmic rays (GCR), solar irradiation, and solar wind (e.g. White et al., 1997;



Zieger and Mursula , 1998; Scafetta and West , 2005; Miyahara et al. , 2008; Thomas et al. , 2013).

Because of this large number of influences and possible interactions the analysis of the temperatures is not easy to interpret, but due to the different timescales of the variations the different types of influences and phenomena can be distinguished sometimes. In this paper we focus on the long term variations of the mesopause temperature with timescales larger than 2 years.

We use OH$^*$ temperatures, which have been derived from groundbased measurements of infrared emissions at a station in Wuppertal (Germany) for our analyses.

The paper is structured as follows. In Sect. 2 we describe the instrument, the measurement technique, and show the OH$^*$ temperature observations, Sect. 3 introduces the Lomb-Scargle periodogram and its properties, and in Sect. 4 we analyse the OH$^*$ temperatures regarding solar correlations, long term trends, and long periodic as well as multi-annual oscillations. A discussion

of the obtained results is given in Sect. 5 and we summarise and conclude in Sect. 6.

## 2   Observations

### 2.1   Instrument and measurements

Excited hydroxyl (OH$^*$) molecules in the upper mesosphere/mesopause region emit radiation in the visible and near infrared. The emission layer is located at about 87 km height with a layer thickness of approximately 9 km (full width at half maximum)

(e.g. Baker and Stair , 1998; Oberheide et al., 2006). The GRIPS-II (GRound based Infrared P-branch Spectrometer) instrument is a Czerny-Turner spectrometer with a Ge detector cooled by liquid nitrogen. It measures the emissions of the P1(2), P1(3), and P1(4) lines of the OH$^*$(3,1) band in the near infrared (1.524 μm–1.543 mum) (for extensive instrument description see Bittner et al., 2000, 2002). The measurements are taken from Wuppertal (51° N, 7° E) every night with a time resolution of about 2 minutes. Thus, a continuous data series throughout a year is obtained with data gaps caused by cloudy conditions

only. This results in approximately 220 nights of measurements per year (Oberheide et al., 2006; Offermann et al., 2010). The relative intensities of the three lines are used to derive rotational temperatures in the region of the OH$^*$ emission layer (see Bittner et al., 2000, and references therein).

At the beginning of 2011 a newly build instrument was operated next to the GRIPS-II instrument. Simultaneous measurements conducted over a few months showed no significant differences between the two instruments. Unfortunately a detector failure

stopped the GRIPS-II measurements mid of 2011, but the new instrument was able to continue the time series of nightly OH$^*$ temperatures. Unfortunately, the new instrument had several technical problems in the following time which led to larger data gaps in the years 2012 and 2013. Finally, a reconstruction was performed to set up the GRIPS-N instrument, a Czerny-Turner spectrometer, equipped with a thermoelectrically cooled InGaAs detector. The optical and spectral properties of GRIPS-N and GRIPS-II are very similar and, thus, the measurements of both instruments are nearly identical. The new GRIPS-N instrument

was operated without further problems since begin of 2014. Hence, for the years 2014 and 2015 a complete set of measurements is available with only the typical data gaps due to cloudiness.





## 2.2 Data processing

The nightly average OH* temperatures derived from the GRIPS-II and GRIPS-N measurements in Wuppertal are shown in the upper panel of Fig.1 for the time interval 1988 to 2015. As mentioned above the data series show larger gaps of several months due to technical problems in the years 2012 and 2013 and, additionally, a data gap of 3 months at the beginning of 1990. These years have to be excluded from the analysis, since a reasonable determination of an annual average temperature in presence of such large data gaps is not possible.

The by far largest variation in this temperature series is the variation in the course of a year. In order to evaluate the data with respect to long term dynamics with periods well above one year the seasonal variation has to be eliminated first. Since the temperature series exhibits data gaps mostly due to cloudy conditions, a simple arithmetic mean for each year is not advisible. We follow the method as used before in several analyses (e.g. Bittner et al., 2002; Offermann et al., 2004, 2006, 2010; Perminov et al. , 2014) and perform a harmonic analysis based on least square fits for each year separately. As described in Bittner et al. (2000) the seasonal variation is characterised by an annual, a semi-annual and a ter-annual cycle. Thus, the temperature variation during one year is described by

$$T = T_0 + \sum_{i=1}^{3} A_i \cdot \sin(\frac{2 \cdot \pi \cdot i}{365.25}(t + \phi_i)), \tag{1}$$

where $T_0$ is the annual average temperature, $t$ is the time in days of year, and $A_i$, $\phi_i$ are the amplitudes and phases of the sinusoids. By fitting this equation to the temperature data we can obtain the best possible estimate of the annual average temperature $T_0$ for each year. A year in this case denotes a calendar year. The resulting annual average temperatures are shown in the lower panel of Fig. 1 with data gaps in the years 1990, 2012 and 2013 (illustrated by the dashed lines). The seasonal variation of the year 2009 is shown in Fig. 2 as a typical example. As described above a detector failure in mid of 2011 stopped the GRIPS-II measurements. The following measurements were performed with a new instrument. The first year of full data coverage with GRIPS-N was 2014. Due to this the corresponding $T_0$ for 2011 and 2014–2015 are marked in red in Fig. 1.

## 2.3 Comparison with other observations

Since there is a data gap of two years (2012–2013) in the GRIPS-II and GRIPS-N measurements in Wuppertal and the last data points are derived from measurements by a new instrument, one has to ensure that the $T_0$ from 2011 to 2015 fit the whole picture of the long term temperature evolution. We compare the Wuppertal observations with observations of OH* temperatures taken from Hohenpeissenberg (48° N, 11° E) to check upon this. The instrument GRIPS-I in Hohenpeissenberg measures in the same spectral range and uses the same data processing technique to determine OH* temperatures. GRIPS-I is an Ebert-Fastie spectrometer with a liquid nitrogen cooled Ge detector (see e.g. Bittner et al., 2002). The measurements at Hohenpeissenberg started end of 2003.

Figure 3 shows the comparison for the two measurement stations. A significant correlation between the two time series can be found with a correlation coefficient r = 0.72. The comparably low value of r is caused by the differences between 2007 to 2009, where the temperatures at Wuppertal partly decrease (increase) and the Hohenpeissenberg temperatures increase (decrease)





at the same time. The linear increase for each time series is shown in Fig. 3 as dashed line in black and red, respectively. In order to get the most appropriate comparison the linear fit to the Hohenpeissenberg time series only considers data points at times where measurements at Wuppertal are also available. The linear increase during the last 12 years at Wuppertal is $(0.46 \pm 0.17)\,\mathrm{K\,year^{-1}}$ and the increase at Hohenpeissenberg is $(0.42 \pm 0.16)\,\mathrm{K\,year^{-1}}$. Both values agree very well, but

the two lines are shifted towards each other indicating an offset between the two stations. This offset is about 0.9 K with Hohenpeissenberg being warmer. In a former study Offermann et al. (2010) obtained a mean offset between the two stations of 0.8 K for the time intervall 2004–2008. Thus, this comparison agrees well the former study. Offermann et al. (2010) suggested the latitudinal difference between the stations to be responsible for this small difference. The temperature differences between the minima 2006 and the maxima 2014 also agree very well for both stations. The values are $(7.3 \pm 0.7)\,\mathrm{K}$ at Wuppertal and

$(6.4 \pm 0.7)\,\mathrm{K}$ at Hohenpeissenberg. Since we analyse the relative evolution of the temperature series at Wuppertal the last data points fit the whole picture of the long term development of OH$^*$ temperatures. Thus, the temperature increase observed at Wuppertal in the recent years is reliable and confirmed by the temperature increase observed at Hohenpeissenberg.

The latest analysis of the OH$^*$ temperatures at Wuppertal regarding long term dynamics was performed for the time interval 1988–2008 (Offermann et al., 2010). The current study now considers a 7 year longer time series until 2015. The clear

temperature increase during the last years has encouraged us to perform a new analysis regarding the long term dynamics.

## 3   Lomb-Scargle periodogram and false alarm probability

Analysing periodicities in the time series of $T_0$ using the common Fast-Fourier-Transform (FFT) or wavelet analysis is not possible, since the time series exhibits data gaps and these methods rely on equidistant data. A frequently used method in such a situation is the Lomb-Scargle periodogram (LSP), which can handle time series with unevenly spacing. The periodogram

was developed by Lomb  (1976) and Scargle  (1982) and is equivalent to the fitting of sinusoids (Horne et al. , 1986). It can be calculated for every frequency $f$, which is another advantage compared to the discrete FFT, which is evaluated at discrete frequencies only. We use the algorithm by Townsend  (2010) for the fast calculation of the periodogram.

An important quantity for the interpretation of a LSP is the so called false alarm probability (FAP). The FAP gives the probability that a peak of height $z$ in the periodogram is caused just by chance, e.g. is caused by noise. As already pointed out by

25 Scargle  (1982), the cumulative distribution function (CDF) can be used to determine the FAP. If we take different samples of noise, calculate the LSP for each sample and then determine the height $z$ of the maximum peak, the CDF of all these heights $z$ gives the probability that there is a height $Z$ smaller or equal to $z$. Consequently the value 1 - CDF gives the probability that there is a height $Z$ larger than $z$ by chance. Thus, 1 - CDF gives the FAP. Another important point in this context is the normalisation of the periodogram, since the normalisation affects the type of distribution of the periodogram and, thus, the

30 description of the FAP (for a more detailed discussion see e.g. Horne et al. , 1986; Schwarzenberg-Czerny , 1998; Cumming et al. , 1999; Zechmeister and Kürster , 2009). We use the normalisation by the total variance of the data, which leads to a beta distribution in the case of gaussian noise (Schwarzenberg-Czerny , 1998). The FAP can then be described by

$$FAP = 1 - [1 - (\frac{2z}{N})^{(N-3)/2}]^{N_i}, \tag{2}$$





where $N$ is the number of data points and $N_i$ is the number of independent frequencies (cf. Schwarzenberg-Czerny , 1998; Townsend , 2010). The number of independent frequencies $N_i$ has to be determined using simulations, since it is not possible to easily describe this quantity analytically (Cumming et al. , 1999). It depends on several factors, e.g. the number of data points $N$ and the spacing of the data points. Horne et al. (1986) showed the partly large effect of the spacing (randomly or clumps of points) on $N_i$. Therefore, we perform simulations to determine $N_i$ for the special situation of our observations. We take random values from a gaussian distribution with mean zero and variance one and the spacing of our observations as input. Then we calculate the LSP for ten thousand of such noise samples and determine the height $z$ of the maximum peak for each LSP. Every LSP is evaluated in the frequency range from Nyquist-frequeny $f = 1/2$ year$^{-1}$ to $f = 1/T$ year$^{-1}$, where $T$ in our case is 35 years, since we want to search for periodicities in range of the time window of the data series of 28 years. Periodicities in this range surely are accompanied with larger uncertainties, but the LSP gives a reasonable overview over the periodicities, even the large ones, included in the time series. The LSP is calculated at $4T_{dur}\Delta f = 53$ evenly spaced frequencies in the mentioned frequency range, where $T_{dur}$ is the duration of observations. Cumming et al. (1999) pointed out that this is an adequate sampling to observe all possible peaks. The upper panel of Fig. 4 shows the resulting empirical CDF of $z$ for our sampling. The number of data points in this case is $N = 25$ and the data series includes the data gaps in 1990 and 2012–2013. The lower panel of Fig. 4 displays the FAP (1 - CDF) as black curve. The fit of the theoretical curve using Eq. 2 to this data points is shown in red. The fit leads to a number of independent frequencies $N_i = 32.4$. With knowledge of $N_i$ we can calculate the FAP for every peak height $z$ and determine confidence levels for the LSP.

## 4 Analysis of long term dynamics: linear trend, solar correlations, long periodic and multi-annual oscillations

### 4.1 Linear trend and 11-year solar cycle

We analyse the long term trend and the correlation with the 11-year cycle in solar activity by means of a multiple linear regression. For this and the following analyses the time coordinate is shifted such as the first data point (1988.5) is set to zero. The annual average temperatures are described by

$$T_0(t, SF) = C_{trend} \cdot t + C_{solar} \cdot SF + b, \tag{3}$$

where $C_{trend}$ and $C_{solar}$ are the two regression coefficients, $t$ is the time in years, $b$ is a constant offset, and $SF$ is the solar radio flux F10.7cm in solar flux units (SFU). The solar radio flux is shown in Fig. 5 for the time interval from 1988 to 2015. There are three solar maxima in this time interval at about 1991, 2001 and 2014. This corresponds well to the annual average temperatures $T_0$, which also show local maxima at these points. The calculated regression coefficients determined by fitting Eq. 3 using the method of ordinary least squares are $C_{trend} = (- 0.089 \pm 0.055)$ K year$^{-1}$ and $C_{solar} = (4.2 \pm 0.9)$ K $(100\,\text{SFU})^{-1}$. The p-values (for the null hypothesis test) are 0.12 for $C_{trend}$ and below 0.01 for $C_{solar}$. The whole fit has a $r^2 = 0.6$. Figure 6 shows the results for this analysis. The upper panel of the figure shows the temperature time series in black and the fit according to Eq. 3 in red. Additionally, the residual $T_{res}$ is shown in the lower panel. Obviously, a fit taking into account a linear trend and the correlation with the 11-year solar cycle is a relatively poor fit to the temperature time series. The temperature residual





still shows a temperature decrease until about 2005 and a temperature increase afterwards. Especially, the large increase at the end of the time series is not captured by the fit. Although there is an increase in solar activity in the same time interval, it is by far not enough to completely explain the observed temperature increase until 2015.

The obvious differences between fit and data series can also be seen in the LSPs in Fig. 7. The LSP is used here to analyse at which periods the determined fit reduces the variance of the original data series. The periodogram for the annual average temperatures $T_0$ is shown in black and the periodogram for the residual $T_{res}$ after subtracting the fit is shown in red. The peak at about 11 years in the LSP for $T_0$, which indicates the correlation with the 11-year solar cycle, disappeared after substracting the fit. In contrast the large broad peak at the end of the periodogram is not completely removed and a peak at a period of about 20 years remains. Please note here, that the periodograms are normalised by the variance of the original time series and the residual, respectively. For the LSP for the residual this means, that the periodicities found and the corresponding false alarm probabilities are valid under the assumption that the fit used to build the residual is "correct" (see e.g. Horne et al. , 1986). The clear signal in the periodogram shows that the fit determined by using Eq. 3 is not sufficient to remove all long term variations. There are two possibilities to describe the long term variation of the temperature series in a better way. Firstly, one can introduce a trend break so that there is a linear decrease in the first part and a linear increase in the second part of the series. Secondly, one can use a long periodic oscillation, which can introduce a trend break with a smoother transition. We will investigate these two possibilities in the next subsections.

## 4.2 Trend break

The trend break and the correlation with the 11-year solar cycle are analysed as follows. In a first step we determine the correlation with the 11-year solar cycle by a linear regression between the temperature time series and the solar radio flux. Thus, we remove the term $C_{trend} \cdot t$ from Eq. 3. The regression coefficient is $C_{solar} = (4.8 \pm 0.9) \, \mathrm{K} \, (100 \, \mathrm{SFU})^{-1}$ (p-value < 0.01).

In the second step we detrend the time series with respect to the 11-year solar cycle and fit two lines to the temperature residual to introduce the trend break. Since the year of the trend break, hereafter denoted as break point, is not known in advance, we use a variable break point and determine the best estimate by means of a least square fit. The residual after eliminating the 11-year cycle is shown in Fig. 8 in black. This residual is described by

$$T_{res}(t) = \begin{cases} C_{trend1} \cdot t + b_1 & : t \le BP \\ C_{trend2} \cdot t + b_2 & : t > BP \end{cases} , \tag{4}$$

where $BP$ is the break point (in years). Since the two different lines need to be equal at the break point, this leads to the condition

$$\begin{aligned} C_{trend1} \cdot BP + b_1 &= C_{trend2} \cdot BP + b_2 \\ \Leftrightarrow b_2 &= b_1 + (C_{trend1} - C_{trend2}) \cdot BP \end{aligned} \tag{5}$$





Thus, Eq. 4 can be rewritten as

$$T_{res}(t) = \begin{cases} C_{trend1} \cdot t + b_1 & : t \leq BP \\ C_{trend2} \cdot t + (b_1 + (C_{trend1} - C_{trend2}) \cdot BP) & : t > BP \end{cases}. \tag{6}$$

The description of the concept and the condition can be seen in Ryan and Porth (2007). Equation 6 now describes the temperature residual for two different phases, where both phases have a linear temperature behaviour. These two phases are coupled

by the variable break point $BP$. We determine the best estimates for the parameters $C_{trend1}$, $C_{trend2}$, $b_1$, and $BP$ by means of a least square fit.

The best fit is shown in Fig. 8 as red line. Additionally, the position of the break point and the corresponding uncertainties are marked as vertical black line and vertical dashed black lines, respectively. We observe a trend break in the middle of year 2006 ($BP$ = (2006.7 ± 2.4) year). Before the trend break in 2006 there is a negative temperature trend $C_{trend1}$ = (-

0.22 ± 0.08) K year$^{-1}$ and after the break point the trend is positive with a slope $C_{trend2}$ = (0.38 ± 0.23) K year$^{-1}$. These findings are also in good agreement with the observations at Hohenpeissenberg, where the temperature series (not corrected for 11-year solar cycle) also shows a minimum at date 2006.5 and an increase afterwards (compare Fig. 3).

### 4.3   Long term oscillation

We analyse the possibility of an oscillation instead of a trend break. Thus, we fit a sinusoid of the form

$$T_{res}(t) = A \cdot \sin(\frac{2 \cdot \pi}{P}(t + \phi)) + b \tag{7}$$

to the temperature residual. $A$ denotes the amplitude, $P$ the period, and $\phi$ the phase. Additionally, we fit an offset $b$, since the mean of the temperature residual is not necessarily identical with the zero crossing of the oscillation. The resulting oscillation is shown in Fig. 8 as blue curve. The important estimated parameters of the fit are an amplitude $A$ = (1.89 ± 0.43) K and a period of about 24 years ($P$ = (23.6 ± 2.6) years). Obviously, this oscillation and the fit using the two linear phases and a

trend break (red lines in Fig. 8) are nearly identical for the time interval after 2006. Before 2006 the blue curve oscillates about the red line. Additionally, the oscillation introduces a much smoother transition from decreasing to increasing temperatures. The decrease in variance is larger for the oscillation than for the fit using two linear phases. The variances of the two resulting differences, $T_{res}$ minus linear trends (red lines) and oscillation (blue curve), respectively, are 2.4 K$^2$ and 1.9 K$^2$. Thus, the oscillation describes $T_{res}$ better, especially at the beginning of the time series. Offermann et al. (2010) already suggested a trend

break in the temperature series at about 1997 and the oscillation accounts for such a second trend break in the temperature series in the mid nineties.

Very prominent is the fact, that the oscillation has a period of about 24 years with a minimum at about 2005 and a maximum at about 1993/1994. This type of oscillation with very similar parameters can be found on the sun. The original solar cycle (Hale cycle) is a cycle with a period of about 22 years and describes the reversal of the magnetic field of the sun. The solar polar mag-

netic field of the sun is shown in Fig. 8 as green curve with a second axis to the right. Evidently, the oscillation fitted to $T_{res}$ and the Hale cycle of the magnetic field are very similar in the time interval shown. The correlation coefficient for a linear regression between the magnetic field and the temperature residual is r = 0.6. The corresponding slope is (1.8 ± 0.5) K (100 µT)$^{-1}$



(p-value $< 0.01$). This is a remarkable accordance between the observed oscillation in atmospheric temperature and solar polar magnetic field.

The long period oscillation describes the largest part of the temperature variability after detrending the temperature series with respect to the 11-year solar cycle. Thus, we analyse the temperature series $T_0$ by means of a multiple linear regression again and include the solar polar magnetic field in the equation, which replaces the linear trend. Hence, Eq. 3 transforms to

$$T_0(SF, B_{solar}) = C_{solar} \cdot SF + C_{hale} \cdot B_{solar} + b, \tag{8}$$

where $B_{solar}$ denotes the solar polar magnetic field and $C_{hale}$ the corresponding regression coefficient. The analysis leads to the results for the regression coefficients $C_{solar} = (5.0 \pm 0.7)\,\mathrm{K}\,(100\,\mathrm{SFU})^{-1}$ and $C_{hale} = (1.8 \pm 0.5)\,\mathrm{K}\,(100\,\mathrm{\mu T})^{-1}$. Both regression coefficients are significant with corresponding p-values below 0.01 and they agree within the uncertainties with the coefficients, which are determined one after the other as shown above. The fit to the temperature time series has a $r^2 = 0.71$. This value is larger than the value for the fit including the 11-year solar cycle and one linear trend, which has a $r^2 = 0.6$ (see Sect. 4.1). Hence, the new description with the solar polar magnetic field as parameter explains more variance of the temperature series. The resulting fit and the residual are shown in Fig. 9. The fit curve (red colour) shows good agreement with the long term variation of the temperature (black dots), but there are still some obvious differences, especially at the beginning of the time series. Additionally, the temperature residual (lower panel of Fig. 9) seems to show a long periodic oscillation. This is confirmed by the LSP for the residual, which is shown in Fig. 10. A small peak at the right end of the periodogram remains. There are two possibilities that explain this remaining peak. Firstly, there is a second oscillation besides the Hale cycle. Secondly, and this is the most likely explanation, there is only one oscillation with a period similar to the Hale cycle but phase shifted. This explanation is supported by the fact, that the oscillation fitted to the temperature residual after substracting the 11-year solar cycle (blue curve in Fig. 8) is slightly phase shifted to the Hale cycle (green curve in Fig. 8). Both the maximum and the minimum of the fitted sinusoid occur somewhat before the extrema in the solar polar magnetic field.

We analyse the second possibility and add an oscillation with a fixed period of 24 years (equal to the period of the fit to the temperature residual) to the multiple linear regression, which replaces the solar polar magnetic field. The equation transforms to

$$T_0(SF, t) = C_{solar} \cdot SF + C_{sin} \cdot \sin(\frac{2 \cdot \pi}{24} \cdot t) + C_{cos} \cdot \cos(\frac{2 \cdot \pi}{24} \cdot t) + b, \tag{9}$$

where $C_{sin}$ and $C_{cos}$ are the amplitudes of the oscillations and $t$ is the time in years. The sum of the sine wave and the cosine wave with fixed period and phases but free amplitudes is one sinusoid with period of 24 years and a free phase ($A \cdot \sin(\frac{2 \cdot \pi}{24} \cdot t + \phi)$). The resulting regression coefficients are $C_{solar} = (4.3 \pm 0.7)\,\mathrm{K}\,(100\,\mathrm{SFU})^{-1}$ (p-value $< 0.01$) for the sensitivity to the solar activity, and $C_{sin} = (1.92 \pm 0.43)\,\mathrm{K}$, and $C_{cos} = (0.34 \pm 0.42)\,\mathrm{K}$ for the amplitudes of the sine and cosine wave. Thus, the amplitude of the 24-year oscillation is $A = (1.95 \pm 0.43)\,\mathrm{K}$ ($A = \sqrt{C_{sin}^2 + C_{cos}^2}$). The fit has a $r^2 = 0.77$, which is an substantial increase. The fit and the residual are shown in Fig. 11. The temperature residual (lower panel of Fig. 11) shows no obvious long term variation any more, neither a linear trend nor an oscillation. Only some variations with periods on the order of several years remain. The LSP for the temperature residual, which is shown in Fig. 12, confirms this. All long term





variations with periods larger than about 10 years are now removed from the temperature series. There are only peaks in the range up to a period of about 8 years. Thus, the description of the annual average temperature including the 11-year solar cycle and an oscillation with a period of 24 years is sufficient to explain all long term variations. No further linear trend can be found in the data series.

## 4.4 Multi-annual oscillations

We briefly analyse the oscillations that remain after substracting the long term variations. These oscillations have periods in the range of several years and we therefore denote them as multi-annual oscillations (MAO). The LSP for the temperature residual (see red curve in Fig. 12) shows three distinct peaks. The first peak is located at a period of about 2.7 years, the second peak is located at about 3.1 years, and the last pronounced peak belongs to a period of about 4.5 years. The FAPs for all peaks is very high with values of slightly below 50% or even higher. But the determination of the FAP (see Sect. 3) is based only on the maximum peak found in a periodogram calculated for noise and there is no additional consideration of other peaks found in the periodogram, which may have a similar height as the maximum peak. If you have a superposition of several sinusoids each with same or similar amplitude, the FAP for one of the corresponding peaks will increase with increasing number of sinusoids. As an example, the height of the peak in a LSP for a single sinusoid is N/2, with N as number of data points (see e.g. Horne et al. , 1986). If you add other sinusoids with same amplitude but different period, the heights of the peaks in the LSP will decrease. Taken 25 data points, as it is the case for the GRIPS measurements, the height of the peak for a single sinusoid is 12.5, for two sinusoids (periods e.g.: 3.0 and 5.0 years) the peak heights of both peaks are approximately 6.5 ,and for three sinusoids (periods e.g.: 3.0, 5.0, and 7.0 years) the peak heights are only about 4.5. The peak height additionally depends on the phases of the different sinusoids, and therefore on the different superposition. Thus, the numbers given are approximations. Nevertheless, the general decrease in height with increasing number of sinusoids is obvious. As a result the FAP for such peaks detected for a superposition of sinusoids would be high, although all oscillations are real. In the case of the temperature residual and the resulting LSP (see red curve in Fig. 9) you get a similar picture. There are several peaks with similar height, which could be caused by a superposition of sinusoids with similar amplitudes, although the FAP for a single peak is high.

We use harmonic fits to get estimates for the three identified MAOs in the temperature residual after removing the 11-year cycle and the 24-year oscillation. A harmonic fit to the temperature residual initiated at a period of 2.7 years leads to the results $P = (2.69 \pm 0.06)$ years and $A = (1.03 \pm 0.33)$ K, a harmonic fit initiated at a period of 3.1 years results in the parameters $P = (3.15 \pm 0.07)$ years and $A = (1.03 \pm 0.33)$ K, and a harmonic fit initiated at a period of 4.5 years results in the parameters $P = (4.54 \pm 0.17)$ years and $A = (0.91 \pm 0.36)$ K. Thus, the amplitudes of all three oscillations are very similar. This would explain the observed LSP, where we can see clear peaks, but the FAP of each peak is very high. As explained above this behaviour is expected for a superposition of several sinusoids. Furthermore, if one subtracts one of the fitted oscillations from the temperature residual and calculates the LSP for the resulting difference, the peak corresponding to the subtracted oscillation is removed, whereas the two other peaks remain. Since this holds for all combinations, we believe that all three MAOs found in the data are real and no artefacts or leakage effects. These three MAOs are the main part of the temperature variance after eliminating the long term variations. But in comparison to the long term variations the observed amplitudes of the MAOs



are much smaller, by about a factor of two. Hence, the whole observed temperature variation is dominated by the long term variations and the MAOs have only minor contribution.

## 5 Discussion

### 5.1 11-year solar cycle

There are numerous publications about the correlation of the 11-year cycle of solar activity and temperatures in the mesopause region. A review is given by Beig (2011a, see Fig. 2 and corresponding section). The sensitivity to the solar activity in the northern mid- to high-latitudes reported in this review is about $1$–$6 \, \mathrm{K} \, (100 \, \mathrm{SFU})^{-1}$. In a more recent study on mesopause temperatures measured at Zvenigorod ($56°$ N, $37°$ E; 2000–2012) by Perminov et al. (2014) a sensitivity of $(3.5 \pm 0.8) \, \mathrm{K} \, (100 \, \mathrm{SFU})^{-1}$ is found. This value perfectly agrees with the result of a former analysis of the GRIPS measurements at Wuppertal (1988–2008),

where also a sensitivity of $(3.5 \pm 0.2) \, \mathrm{K} \, (100 \, \mathrm{SFU})^{-1}$ was found (Offermann et al., 2010). In our study we obtained results in the range between $4$–$5 \, \mathrm{K} \, (100 \, \mathrm{SFU})^{-1}$. Depending on the analysis method the results slightly differ from each other, but they all agree within the uncertainties. Since the parameters for the multiple linear regressions (solar radio flux, solar polar magnetic field, 24-year oscillation, and time) are not completely independent of each other, the derived regression coefficients are good approximations to the "true" values. Much longer time series including more solar maxima would be necessary to

finally derive the "true" regression coefficients. Thus, small differences in the derived values are expected, especially in the case of the multiple linear regression including the solar radio flux and the linear trend, since this regression leads to a result that cannot completely explain all long term trends and oscillations in the time series. All derived values for the sensitivity of the OH* temperatures to the 11-year solar cycle are slightly larger than the one derived in the former analysis of the GRIPS measurements at Wuppertal. But the time intervals are different for the analyses, which can lead to different results for the

derived sensitivities. This aspect was already discussed by Offermann et al. (2010).

Although the derived values are in the expected range for northern mid- to high-latitudes, one new aspect with respect to the correlation between 11-year solar cycle and mesopause temperatures has become apparent. In the present study the correlation was determined for three solar maxima including the comparably weak latest solar cycle 24. Our study shows that the significant correlation between OH* temperatures and the 11-year solar cycle is still evident in this case.

### 5.2 Linear trend and trend break

Temperature trends in the mesopause region are reported in a number of papers, and a review about numerous results is given by Beig (2011b, see Fig. 2 and corresponding section). The temperature trends reported there range between no trend up to a cooling of about $3 \, \mathrm{K \, decade}^{-1}$. The recent analysis by Perminov et al. (2014) for the measurements at Zvenigorod ($56°$ N, $37°$ E) showed a trend of $(-2.2 \pm 0.9) \, \mathrm{K \, decade}^{-1}$ for the time interval 2000–2012. In a former study of the Wupper-

tal OH* temperature series (1988–2008) a negative trend of $(-2.3 \pm 0.6) \, \mathrm{K \, decade}^{-1}$ was found (Offermann et al., 2010). The multiple linear regression using the solar radio flux and time as parameters in this paper results in a cooling trend of





(-0.89 ± 0.55) K decade$^{-1}$ for the Wuppertal OH$^*$ temperatures from 1988 to 2015 (see Sect. 4.1). This value is significantly smaller than the trend derived in the former study of the Wuppertal data. Since there is an increase in temperature since about 2006 and the former study by Offermann et al. (2010) ended 2008, this temperature increase leads to a smaller negative trend in our study. But as shown above one linear trend is not sufficient to acount for all long term variation in the time series. Due

to this we introduced a trend break and found a negative trend before year 2006 and a positive trend afterwards. The obtained values are (-2.2 ± 0.8) K decade$^{-1}$ and (3.8 ± 2.3) K decade$^{-1}$, respectively (see Sect. 4.2). The time interval used in the former study of the Wuppertal OH$^*$ temperature series by Offermann et al. (2010) is nearly identical with the time interval of the first phase showing the negative temperature trend. The linear temperature trends derived by Offermann et al. (2010) and in this study for this time interval perfectly agree. Due to the additional 7 years of observations this study now clearly shows

that the former negative linear trend turned into a positive trend in the last years.

### 5.3 Long term oscillation

The observed trend break can also be described using a long periodic oscillation. In Sect. 4.3 we show two different possibilities for such a long periodic oscillation.

Firstly, the solar polar magnetic field (Hale cycle) is used as one parameter in a multiple linear regression with the second

parameter being the solar radio flux. The correlation coefficients are $C_{solar}$ = (5.0 ± 0.7) K $(100\,\mathrm{SFU})^{-1}$ and $C_{hale}$ = (1.8 ± 0.5) K $(100\,\mathrm{\mu T})^{-1}$ ($r^2$ = 0.71). But especially at the beginning of the time series the fit curve is not perfectly matching the observations (see Fig.9). Additionally, the LSP for the temperature residual after subtracting this fit curve still shows a peak corresponding to a long period (red curve in Fig. 10). Thus, the Hale cycle together with the 11-year solar cycle cannot explain all observed long term dynamics. Because of these facts, we believe that the solar polar magnetic field as acting input parameter

is not very likely.

Secondly, an oscillation with a period of 24 years is used to describe the OH$^*$ temperature time series. A multiple linear regression using the solar radio flux and this 24-year oscillation leads to the regression coefficient $C_{solar}$ = (4.3 ± 0.7) K $(100\,\mathrm{SFU})^{-1}$ and an amplitude of the oscillation $A$ = (1.95 ± 0.43) K ($r^2$ = 0.77). After subtracting the derived fit curve the LSP for the residual do not show any remaining long periodic signals (see Fig. 12). The 24-year oscillation, shown in Fig. 13 as black

curve (with full circles), is phase shifted compared to the Hale cycle and the extrema occur slightly before the extrema of the solar polar magnetic field (compare Fig. 8 green curve and Fig. 13 black curve; e.g. maximum at about 1993 compared to 1994/1995). This time shift supports the opinion that the Hale cycle is not very likely as an acting input parameter. The nature of the 24-year oscillation is not clear yet, but a selfsustained oscillation in the atmosphere would be a real possibility. Such oscillations were recently discovered by Offermann et al. (2015). An oscillation with a period of about 20 to 25 years is found

in various atmospheric parameters such as temperature (Qu et al. , 2012; Wei et al. , 2015), geopotential height (Coughlin and Tung , 2004a, b), and planetary wave activity (Jarvis , 2006; Höppner and Bittner , 2007). It is also seen in two atmospheric models (HAMMONIA, WACCM). A detailed discussion is, however, beyond the scope of this paper.

The most important point here is that no additional linear trend can be maintained. All long term dynamics of the Wuppertal OH$^*$ temperature time series can be described as a combination of the 11-year solar cycle and a 24-year oscillation. With the





knowledge of this 24-year oscillation the linear trends derived in this study (see Sect. 4.1) and a former study of the Wuppertal OH* temperature time series can be reproduced. Figure 13 demonstrates that very different trends can be calculated if specific time intervals of the (sinusoidal) data are used. By fitting a line to the corresponding part (time interval) of the data we obtain the linear trend. The linear trend for the time interval analysed in this study (1988–2015) is (-0.087 $\pm$ 0.033) K year$^{-1}$, which

is identical to the linear trend $C_{trend}$ = (- 0.089 $\pm$ 0.055) K year$^{-1}$ derived by using a multiple linear regression with time and solar radio flux as parameters (see Sect. 4.1). This linear trend is shown in Fig. 13 as red line (with squares). Offermann et al. (2010) derived a linear trend for the time interval 1988–2008 of (-0.23 $\pm$ 0.06) K year$^{-1}$. A linear fit to the data for this time interval leads to a slope of (-0.21 $\pm$ 0.03) K year$^{-1}$ (green line (with triangles) in Fig. 13). Thus, the 24-year oscillation "explains" the derived linear trends of this and the former study as well as the obvious trend break observed in 2006. This

means that all different kinds of linear trends are possible depending on the time interval which is analysed. If we continue the oscillation back to 1975 (black dashed line in Fig. 13) and fit a line to these "data" for the whole time interval (1975–2015; blue line (with plus signs)) in Fig. 13), this leads to a slope of (0.015 $\pm$ 0.012) K year$^{-1}$. Surely, this continuation is an assumption and cannot be verified by the observations, but it is likely and clearly shows the possible effects. The presence of such a long periodic oscillation that in combination with the 11-year solar cycle explains all long term dynamics without an additional

linear trend is very important with respect to any kind of comparison between different observations or model simulations. Each comparison of linear trends is only valid if the same time interval is analysed. Furthermore, the current study suggests that there is no universal linear trend which is valid for all time intervals at this altitude.

### 5.4    Multi-annual oscillations

After detrending the OH* temperature series regarding the long term variations (11-year solar cycle and 24-year oscilla-

tion) MAOs with periods below 8 years remain. The most prominent oscillations have periods of $P$ = (2.69 $\pm$ 0.06) years, $P$ = (3.15 $\pm$ 0.07), and $P$ = (4.54 $\pm$ 0.17) years with corresponding amplitudes of $A$ = (1.03 $\pm$ 0.33) K, $A$ = (1.03 $\pm$ 0.33) K, and $A$ = (0.91 $\pm$ 0.36) K. respectively. In a recent study we analysed the SABER temperatures in the region 45° N–55° N and 4° W–16° E in the time interval from 2002 to 2012 (see Offermann et al., 2015). Beside oscillations with other periods the SABER temperatures show a MAO with a period of about 2.6 years and an amplitude of about 1 K in altitude range between

80 to 90 km. Thus, there is an good agreement between this MAO and the MAO with period of about 2.69 years found in the GRIPS observations. The study by Offermann et al. (2015) indicates that the MAOs identified in the atmosphere are selfsus-tained oscillations, since they are also present in model simulations by the HAMMONIA model with climatological boundary conditions.

### 6    Summary and conclusions

We present the analysis of the OH* temperatures derived from the GRIPS measurements at Wuppertal. We use annual average temperatures in the time interval 1988 to 2015 for our study. The study mainly focuses on the long term dynamics as well as on MAOs with periods between 2 to 5 years. The analysis leads to the following results:



1. The OH$^*$ temperatures show a significant correlation with the solar radio flux. We find a sensitivity to the 11-year solar cycle of 4–5 K $(100\,\mathrm{SFU})^{-1}$.

2. One linear trend during the whole time interval (together with the sensitivity to the 11-year solar cycle) cannot sufficiently explain all long term dynamics found in the OH$^*$ temperatures. We introduce a trend break to better account for these long term dynamics. The best representation of the temperature series is found if the trend break occurs in mid 2006 (date = $(2006.7 \pm 2.4)$ years). Before the break point the linear trend is negative and after the break point the trend turns positive with the slopes of $(-0.22 \pm 0.08)$ K year$^{-1}$ and $(0.38 \pm 0.23)$ K year$^{-1}$, respectively.

3. The reversal of the temperature trend can also be described as a long periodic oscillation. We present two possibilities for this oscillation. Firstly, the solar polar magnetic field of the sun (Hale cycle) is used in a multiple linear regression together with the solar radio flux as second parameter. The derived regression coefficients are $C_{solar} = (5.0 \pm 0.7)$ K $(100\,\mathrm{SFU})^{-1}$ and $C_{hale} = (1.8 \pm 0.5)$ K $(100\,\mu\mathrm{T})^{-1}$ ($r^2 = 0.71$). Secondly, a 24-year oscillation is used instead of the Hale cycle, which leads to the best description of the OH$^*$ temperatures series, at all. A multiple linear regression leads to the coefficients $C_{solar} = (4.3 \pm 0.7)$ K $(100\,\mathrm{SFU})^{-1}$, $C_{sin} = (1.92 \pm 0.43)$ K, and $C_{cos} = (0.34 \pm 0.42)$ K ($r^2 = 0.77$). The amplitude of the 24-year oscillation is $A = (1.95 \pm 0.43)$ K ($A = \sqrt{C_{sin}^2 + C_{cos}^2}$). The most important point here is that no additional linear trend is needed and that the combination of 24-year oscillation and 11-year solar cycle explains all long term dynamics. This is especially satisfying as the notion "trend break" is somewhat "non-physical".

4. After detrending the temperature series regarding the 11-year solar cycle and the 24-year oscillation MAOs remain. Harmonic fits to the detrended temperature series lead to oscillations with periods of $P = (2.69 \pm 0.06)$ years, $P = (3.15 \pm 0.07)$, and $P = (4.54 \pm 0.17)$ years with corresponding amplitudes of $A = (1.03 \pm 0.33)$ K, $A = (1.03 \pm 0.33)$ K, and $A = (0.91 \pm 0.36)$ K, respectively.

5. A caveat arises when estimating linear trends from data sets containing long term variations. Trend results are quite sensitive to the length of the data interval used. This is especially important for any kind of comparison.

*Acknowledgements.* This work was funded by the German Federal Ministry of Education and Research (BMBF) within the ROMIC (Role Of the Middle atmosphere In Climate) project MALODY (Middle Atmosphere LOng term Dynamics) under Grant no. 01LG1207A. Wilcox Solar Observatory data used in this study was obtained via the web site http://wso.stanford.edu at 2016:04:11_08:31:21 PDT courtesy of J.T. Hoeksema. The Wilcox Solar Observatory is currently supported by NASA. The solar radio flux 10.7cm data was obtained from the Natural Resources Canada, Space Weather Canada website: http://www.spaceweather.gc.ca/.



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





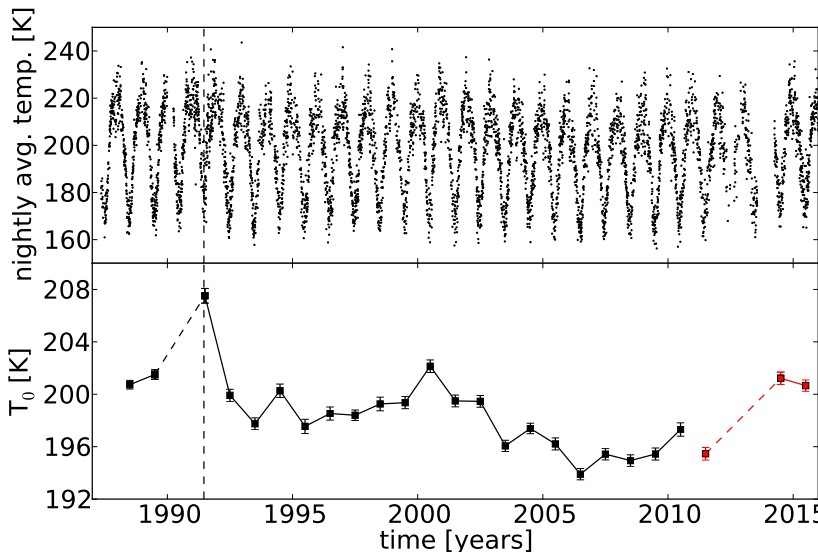

**Figure 1.** OH$^*$ temperature time series derived from GRIPS-II and GRIPS-N measurements at Wuppertal. The upper panel shows the nightly average temperatures and the lower panel shows the annual average temperatures $T_0$. Each $T_0$ is plotted in the middle of the corresponding year and the dates given at the x-axis show the beginning of the years. The annual average temperatures partly or completely derived from the new instrument between 2011 and 2015 are shown in red. The error bars show the estimated standard deviation of the fit parameter. The vertical dashed line marks the date of Mt. Pinatubo eruption.



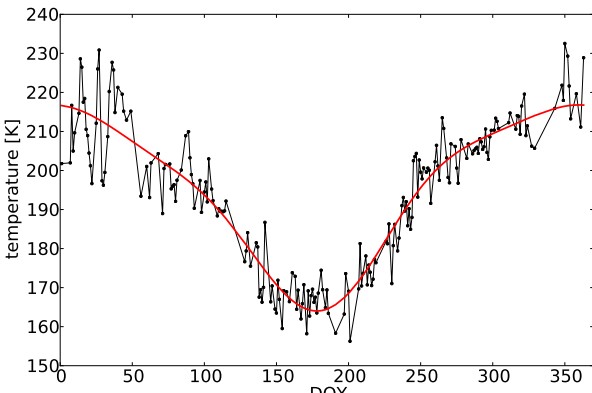

**Figure 2.** GRIPS-II nightly average temperatures of 2009 plotted at the day of year (DOY). The measurement data are shown in black and the harmonic fit using Eq. 1 is shown as the red curve.





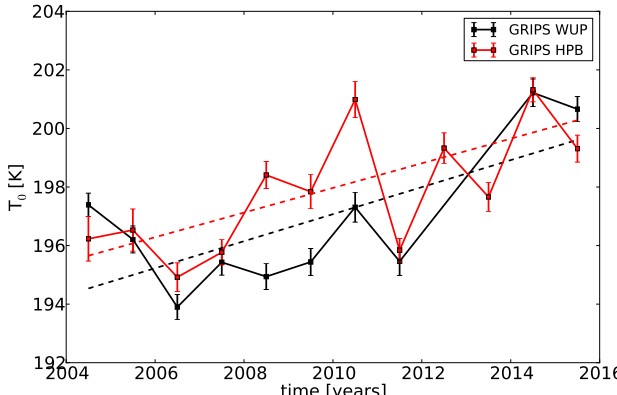

**Figure 3.** OH* annual average temperatures for the two stations Wuppertal and Hohenpeissenberg in the time interval 2004–2015. The temperatures for Wuppertal (WUP) are shown in black and the temperatures for Hohenpeissenberg (HPB) in red. The dashed lines show the linear fits to the corresponding time series. The linear fit for the Hohenpeissenberg time series only considers measurements at times Wuppertal measurements are also available.




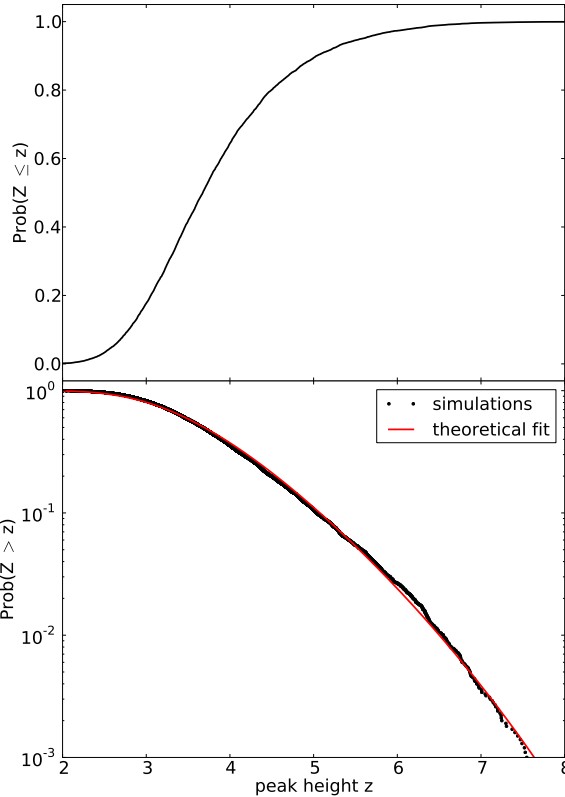

**Figure 4.** Distribution for peak heights $z$ determined using random values from a gaussian distribution as input for the calculation of LSP (for details see Sect. 3). The upper panel shows the empirical CDF, thus, the probability that there is a height $Z$ smaller or equal to $z$. The FAP (probability that a height $Z$ larger $z$ occurs just by chance) is shown in the lower panel. The simulation results are shown in black and a fit to the theoretical curve from Eq. 2 is shown in red. Note the logarithmic scale of the y-axis of the lower panel. This calculations are done for a data sampling same as that of the time series from 1988 to 2015 including data gaps. The fit leads to a number of independent frequencies $N_i = 32.4$.





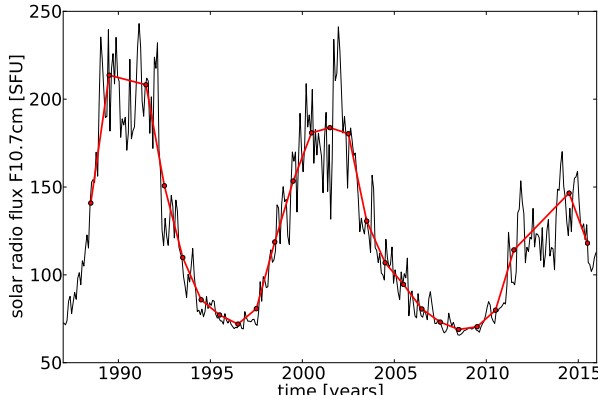

**Figure 5.** Monthly average values of the solar radio flux F10.7cm. The red dots mark the annual average values corresponding to the times of the GRIPS data points. The data were provided by Natural Resources Canada, Space Weather Canada.





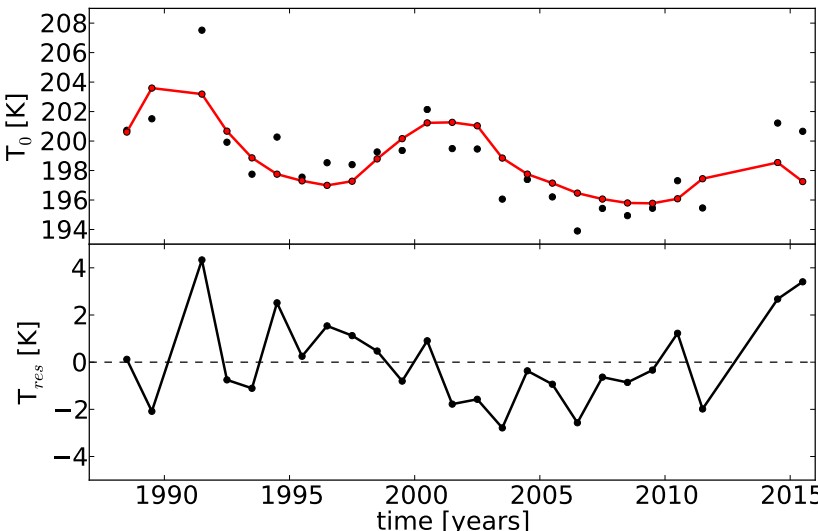

**Figure 6.** The upper panel of the figure shows the time series of annual average OH* temperatures in black and the fit corresponding to Eq. 3 with the regression coefficients $C_{trend}$ = (0.089 ± 0.055) K year$^{-1}$ and $C_{solar}$ = (4.2 ± 0.9) K (100 SFU)$^{-1}$ in red. In the lower panel the residual $T_{res}$ of the two is shown.





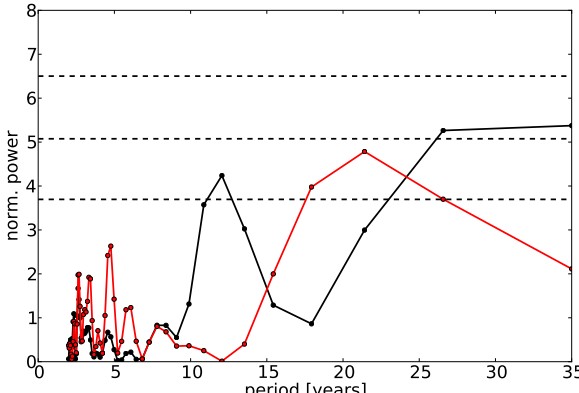

**Figure 7.** The Lomb-Scargle periodogram for the time series of annual OH* temperatures (see Fig. 1 lower panel) is shown in black and the LPS for the residual after subtracting the fit according to Eq. 3 (see Fig. 6 lower panel) is shown in red. The LSP is evaluated at 53 evenly spaced frequencies in the range $f = 1/2$ year$^{-1}$ to $f = 1/35$ year$^{-1}$. The dashed black horizontal lines display the levels for false alarm probabilities of 0.01, 0.1, and 0.5 (top to bottom), respectively. The false alarm probabilities are calculated according to Eq. 2 using $N_i = 32.4$ and the number of data points $N = 25$.



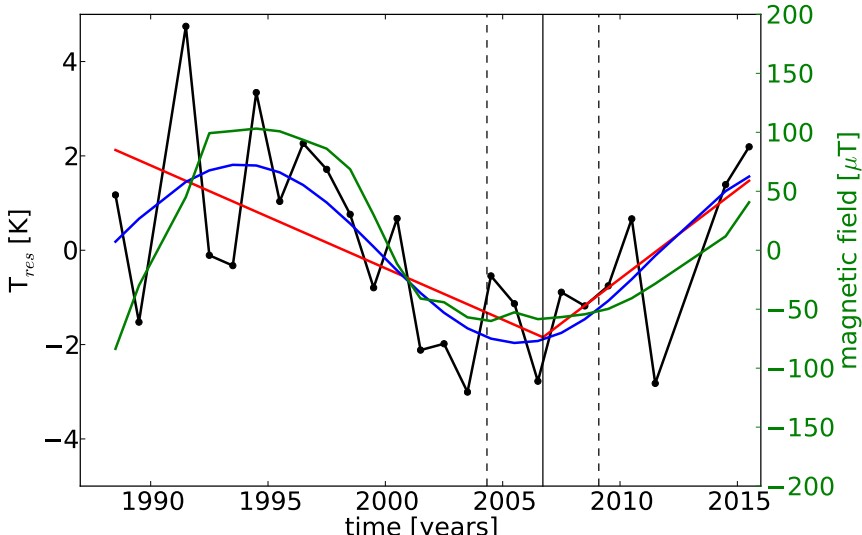

**Figure 8.** Residual for the temperature time series after removing the 11-year solar cycle. The red lines show the fit according to Eq. 6 and the blue curve the fit according to Eq. 7. The break point BP is marked by the vertical black line and the corresponding uncertainties are shown as vertical dashed black lines. Additionally, the solar polar magnetic field is displayed as green curve with a second axis to the right. Shown are the average values for the solar north and south pole with the magnetic field orientation of the north pole. The data were provided by the Wilcox Solar Observatory (for an instrument description see Scherrer et al. , 1977).





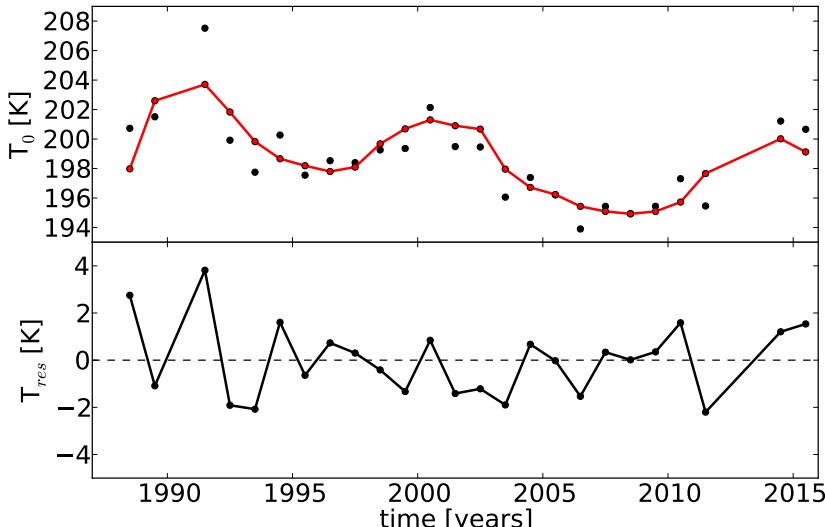

**Figure 9.** The upper panel of the figure shows the time series of annual average OH$^*$ temperatures in black and the fit corresponding to Eq. 8 with the regression coefficients $C_{hale} = (1.8 \pm 0.5)\,\mathrm{K}\,(100\,\mu\mathrm{T})^{-1}$ and $C_{solar} = (5.0 \pm 0.7)\,\mathrm{K}\,(100\,\mathrm{SFU})^{-1}$ in red. In the lower panel the residual $T_{res}$ of the two is shown.





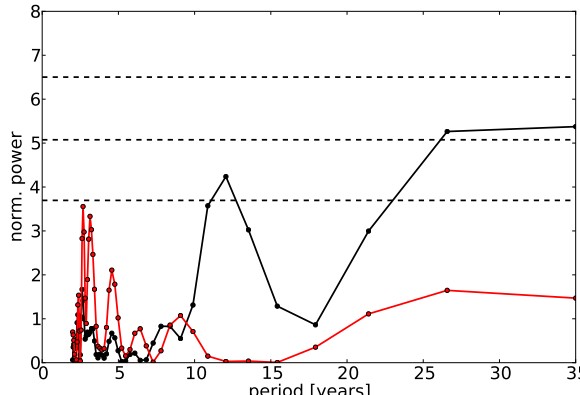

**Figure 10.** The Lomb-Scargle periodogram for the time series of annual OH* temperatures (see Fig. 1 lower panel) is shown in black and the LPS for the residual after subtracting the fit according to Eq. 8 (see Fig. 9 lower panel) is shown in red. For details see description of Fig. 7.



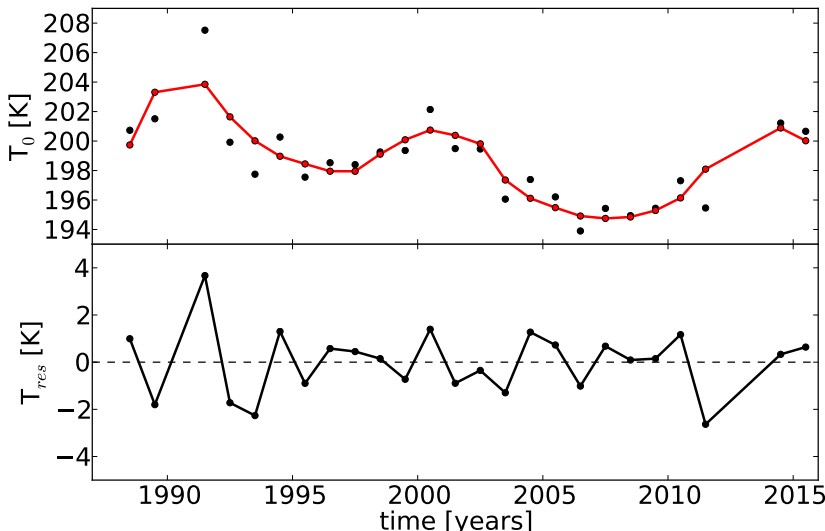

**Figure 11.** The upper panel of the figure shows the time series of annual average OH* temperatures in black and the fit corresponding to Eq. 9 with the regression coefficients $C_{solar} = (4.3 \pm 0.7)\,\mathrm{K}\,(100\,\mathrm{SFU})^{-1}$, $C_{sin} = (1.92 \pm 0.43)\,\mathrm{K}$ and $C_{cos} = (0.34 \pm 0.42)\,\mathrm{K}$ in red. In the lower panel the residual $T_{res}$ of the two is shown.




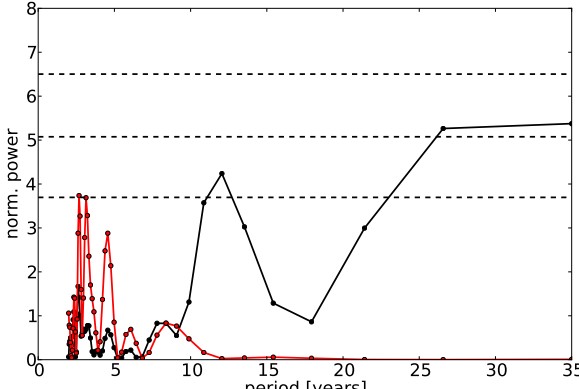

**Figure 12.** The Lomb-Scargle periodogram for the time series of annual OH* temperatures (see Fig. 1 lower panel) is shown in black and the LPS for the residual after subtracting the fit according to Eq. 9 (see Fig. 11 lower panel) is shown in red. For details see description of Fig. 7.





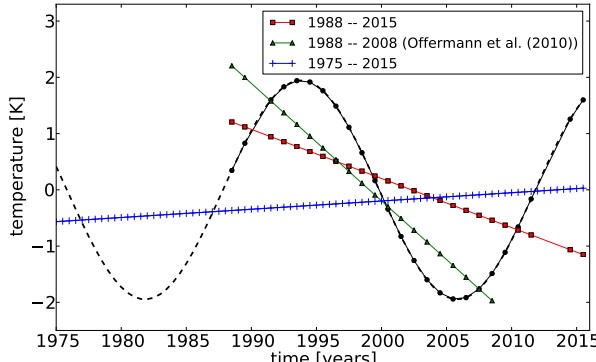

**Figure 13.** 24-year oscillation of OH* temperatures resulting from the multiple linear regression using Eq. 9. The regression coefficients are $C_{sin} = (1.92 \pm 0.43)$ K and $C_{cos} = (0.34 \pm 0.42)$ K. The amplitude of the 24-year oscillation is $A = (1.95 \pm 0.43)$ K ($A = \sqrt{C_{sin}^2 + C_{cos}^2}$). The solid black line (with full circles) shows the oscillation for the analysed time interval 1988–2015 and the dashed black line shows the continuation of this oscillation back to 1975. The red line (with squares) displays a linear fit to the oscillation for the time interval 1988–2015, the green line (with triangles) the fit for the interval 1988–2008, and the blue line (with plus signs) a fit to the interval 1975–2015.