# Peer review of "Long term dynamics of OH\* temperatures over Middle Europe: Trends and solar correlations"

_Atmospheric Chemistry and Physics, 2016_

## Referee Comment (RC1) · Anonymous Referee #2 · 22 Jun 2016

The paper seeks to explore an alternative explanation for the change in slope of the OH temperature time series previously reported in earlier publications using a shorter time series. While the results are promising, I believe there is an overall shortcoming in the analysis technique. While this may not substantially change the overall results, it does put into question their statistical significance.

When fitting data using an orthogonal basis set, one can fit the functions simultaneously or sequentially, where the first function is subtracted from the data and the next fit to the residuals. However, in this paper there is a combination orthogonal functions (sinusoids in the case of the periodogram) and non-orthogonal functions (solar cycles and trends) fitted to the data sequentially. For example in Section 4.1, the solar 10.7 cm flux and a trend first are correctly fitted simultaneously as they are non-orthogonal. However, on page 7, lines 1-13, a periodogram, which fits a set of orthogonal sinusoids, is performed on the data that has had the solar and trend subtracted. It is not technically valid to perform the LSP of the residuals since functions non-orthogonal to the resulting periodogram sine waves have been removed. In particular, on line 5, it is stated that the LSP shows a peak with a period of 20 years remains after subtraction of the solar cycle and trend. However, the subtraction of a linear trend will act as a low pass filter (Kennedy, JGR 85, p219, 1980). Thus, the "peak" at 20 years is likely an artifact of passing the original data with its red-noise through this low pass filter.

Though the paper tries to justify this referring to Horne et al., 1986, it should be noted that Hornes's analysis removed a function orthogonal to the other periodogram components. That is not the case here. Additionally, in Horne's analysis, the estimate of variance after removal of the orthogonal function must be used to re-normalize the periodogram. That estimate of variance must be adjusted to the reduction in the number of degrees of freedom associated with the removal of these "correct" functions in much the same way that the total variance of the raw data set was estimated using N-1 to account for the removal of the mean. Here, a mean, a trend, and a solar cycle (itself a combination of several orthogonal sinusoids) have been removed, and the variance must be estimated appropriately.

In section 4.2, the solar F10.7 cycle is not an orthogonal function to the two linear trends. Hence, its removal before fitting the two trends will influence the trends. All of these non-orthogonal components should be fit simultaneously.

Similarly, in section 4.2, a sinusoid is fitted to the residual after removal on non-orthogonal components. It should be fit simultaneously with the other terms as in Equation 8. Page 9, line 10, it is not mentioned whether this procedure explains more of the variance than the trend-break (assuming the trend break analysis is done correctly).

Page 9, line 13. As previous comments, one has not removed orthogonal functions nor adjusted the estimate of variance to account for the removal of these functions.

Thus the LSP of the residual cannot be interpreted as periodic components remaining in the data. These may well have been created by the removal of the non-orthogonal functions.

Page 9, line 20. Again, the peak at 24 years in the periodogram could well be the result of the non-orthogonal functions being used to fit the data, and does not justify fixing the period to be fixed at 24 years in the fitting procedure. Thus, the period should also be allowed to vary in the fit in equation 9. Otherwise, there is no unambiguous evidence from the periodogram that it is exactly 24 years.

In Section 4.4, , the LSP of the residuals is again not valid (as stated above), and these periods should not be fixed. One could fit a series of sinusoids simultaneously with the solar cycle, where the periods are also treated as free parameters. Using a non-linear least squares fit, the periodogram of the residuals may be used to estimate the initial values of these parameters. However, it is unclear whether there are enough degrees of freedom in the 25 data points to accommodate this analysis.

I would advise that the fitting be performed simultaneously rather than sequentially to avoid the problems associated with the non-orthogonal basis set. In this way, the statistical significance of the periods, and the values of the periods themselves may be assessed. If indeed the results are substantially the same, then the interpretation will hold.

---

## Referee Comment (RC2) · Anonymous Referee #1 · 1 Jul 2016

This paper opens a new view on trends in mesopause temperatures. Since it is controversial in comparison with other results, possible open questions or weaknesses of the paper should be carefully revised. Therefore I recommend major revision but, on the other hand, I expect final publication of the paper even though my personal point of view in its area differs.

Comments:

You assume that dependence of temperature on solar proxies is the same over the whole period of measurements. Some recent results indicate that it need not be true. Try to calculate solar proxy dependence separately for 1988-2005 and 2006-2015, even though the second period is rather short for significant result.

It is natural that eq. (9) with three terms provides better fit than eq. (8) with two terms.

[Figure]

Try to add to eq. (8) trend term as the third term and compare it with results of eq. (9).

I do not agree with your conclusion (5) as it is written. The result that if you assume only one trend over the whole period than the trend depends on the length of period is trivial and well known but it does not mean that linear trend approach is wrong. Since long-term changes of some trend drivers (ozone, geomagnetic activity etc.) are temporally and spatially variable, trends can (and some must) change with time (see e.g. review paper Lastovicka et al. (2012)). For this reason a piecewise linear trend model with trend break(s) has been introduced, which you also apply. So either re-formulate your conclusion (5) or delete it. Lastovicka, J., S.C. Solomon, L. Qian: Trends in the Neutral and Ionized Upper Atmosphere. Space Science Reviews, 168, 113-145, doi: 10.1007/s11214-011-9799-3, 2012.

Page 7: The trend break near 1997 is not well supported by data in Fig. 8; it appears to be rather questionable.

Figure 3: Have you some explanation for WUO-HPB difference in 2008-2010?

Figures 8 and 9: Eye inspection of data points in these Figures calls for trend break rather in 2003 than in 2006, even though both are possible.

You should also take into account results of some other authors:

C.M. Hall et al.: Temperature trends at 90 km over Svalbard, Norway (78 N, 16 E), seen in one decade of meteor radar observations. J. Geophys. Res., 117, D08104, doi: 10.1029/2011JD017028, 2012: No change of 90 km temperature trend over Svalbard around 2006. Meteor radar temperatures.

I.A. Mokhov, A.I. Semenov: Joint analysis of the long-term behavior of temperature in the mesopause and on the Earth's surface during the period of about 60 years. 6th NDMC Meeting, Grainau, 2014 (your co-author D. Offermann attended): 1960-2012 data from Zvenigorod near Moscow. OH mesopause temperatures. Drop in the second half of the 1970s, than negative trend till the mid-1990s and essentially no trend

afterwards. No change reported around 2006.

C.-Y. She, D.A. Krueger, T. Yuan: Long-term midlatitude mesopause region temperature trends deduced from quarter century (1990-2014) Na lidar observations. Ann. Geophysicae, 33, 363-369, doi: 10.5194/angeocom-33-363-2015, 2015. Fort Collins/Logan, USA. 85-90 km temperature trend 1990-2014 is negative, after the mid-1990s weaker than before. No change of trend reported around 2006.
* * *

---

## Author Comment (AC1) · 13 Sep 2016

The comment was uploaded in the form of a supplement:
http://www.atmos-chem-phys-discuss.net/acp-2016-300/acp-2016-300-AC1-
supplement.pdf

---

## Author Response (AR1)

Reply to the reviewer comments on the paper

"Long term dynamics of OH* temperatures over Middle Europe: Trends and solar correlations"

by C.Kalicinsky et al.

We thank the referees for their helpful comments and recommendations. In the following, we discuss the issues addressed by the referees and explain our opinions and the modifications of our manuscript.
We enumerate the referee comments and repeat them in bold face. The modifications of the manuscript are displayed in the marked-up manuscript version as colored text. Deleted parts are shown in red and new text parts in blue.

**1 Comments Referee 1**

**This paper opens a new view on trends in mesopause temperatures. Since it is controversial in comparison with other results, possible open questions or weaknesses of the paper should be carefully revised. Therefore I recommend major revision but, on the other hand, I expect final publication of the paper even though my personal point of view in its area differs.**

1. **You assume that dependence of temperature on solar proxies is the same over the whole period of measurements. Some recent results indicate that it need not be true. Try to calculate solar proxy dependence separately for 1988-2005 and 2006-2015, even though the second period is rather short for significant result.**
   We analysed if the assumption of a constant dependence of the temperature on solar proxies is true. The procedure we used is as follows. We take different time intervals with a length of 11 years, which is approximately the length of one solar cycle. We believe that this is the shortest time interval that can be used to derive significant results. The first time interval used is $1988 - 1998$ and then we shift the interval several times by one year until the end of the whole time period is reached. For each time interval we derive the solar dependence and one linear trend. Here we follow the comments made by referee 2 that the non-orthogonal funtions solar cycle and trend have to be fitted simultaneously. Thus, we use Eq. 3 for the fit. The results are presented in the manuscript in the new Fig. 14. The dependence on the solar proxy does not show significant changes during the whole time period of observations. The mean of the derived sensitivies is $(3.9 \pm 0.3)\,\mathrm{K}\,(100\,\mathrm{SFU})^{-1}$, which agrees with the results obtained by fitting the solar dependence and one oscillation for the whole data set $((4.1 \pm 0.8)\,\mathrm{K}\,(100\,\mathrm{SFU})^{-1})$. The derived trend values show an oscillation with the parameters $A = (0.36 \pm 0.06)\,\mathrm{K}$ for the amplitude and $P = (23.2 \pm 2.5)$ years for the period. This oscillation agrees within the uncertainties with the period and phase of the derivative of the temperature oscillation derived by fitting the

solar dependence and one oscillation for the whole data set. Thus, the analysis by using 11-year time intervals confirms the results obtained by analysing the whole data set at once. In total, the assumption that the dependence of the temperature on solar proxies is the same over the whole period of observations appears to be correct for the Wuppertal OH$^*$ temperatures.

We added new Sects. 4.4 and 5.4 and presented the results of this analysis.

2. **It is natural that eq. (9) with three terms provides better fit than eq. (8) with two terms. Try to add to eq. (8) trend term as the third term and compare it with results of eq. (9).**
We added a trend term to Eq. 8:

$$T_0(SF, B_{solar}, t) = C_{solar} \cdot SF + C_{hale} \cdot B_{solar} + C_{trend} \cdot t + b,$$

The result by using this equation is not siginifically different from the result by using the equation without the trend term. The $r^2$ increases by only 0.001 and, thus, Eq. 9 explains still more variance of the temperature series. Furthermore, the obtained trend is not significantly different from zero. The obtained value is $C_{trend} = (\text{-}0.013 \pm 0.054)$ K year$^{-1}$. Since the obtained trend is not significant, we keep at using the Eq. 8 as it is. We additionally added a sentence to Sect. 4.3 to explain this fact.

3. **I do not agree with your conclusion (5) as it is written. The result that if you assume only one trend over the whole period than the trend depends on the length of period is trivial and well known but it does not mean that linear trend approach is wrong. Since long-term changes of some trend drivers (ozone, geomagnetic activity etc.) are temporally and spatially variable, trends can (and some must) change with time (see e.g. review paper Lastovicka et al. (2012)). For this reason a piecewise linear trend model with trend break(s) has been introduced, which you also apply. So either reformulate your conclusion (5) or delete it. Lastovicka, J., S.C. Solomon, L. Qian: Trends in the Neutral and Ionized Upper Atmosphere. Space Science Reviews, 168, 113-145, doi: 10.1007/s11214-011-9799-3, 2012**

We reformulated our conclusion 5) and we added a sentence to the introduction where we already describe the possibility of trend breaks and the use of piecewise linear trends.

4. **Page 7: The trend break near 1997 is not well supported by data in Fig. 8; it appears to be rather questionable.**

The date 1997 is cited from Offermann et al., 2010. It is correct, that the possible trend break obtained by our analysis seems to occur earlier at about 1993/94. We reformulated the corresponding sentence.

5. **Figure 3: Have you some explanation for WUO-HPB difference in 2008-2010?**
   We believe that the difference is caused by some kind of local effects. The largest difference in 2010 for example is caused by an exeptional warm summer in Hohenpeissenberg. This larger temperatures in summer are also observed at the nearby station in Oberpfaffenhofen (see C.Schmidt et al., 2013, their Fig. 12) but not in Wuppertal. We added a sentence about this fact to Sect. 2.3.

6. **Figures 8 and 9: Eye inspection of data points in these Figures calls for trend break rather in 2003 than in 2006, even though both are possible.**

   Due to the scatter of the data points, different break points seem to be possible. The new refined analysis, where all dependencies are fitted at once, leads to a break point in mid 2008, whereas the minimum of the 25-year oscillation is located at 2005/6. The break point given is the result of the best fit to the data series.

7. **You should also take into account results of some other authors:**
   **C.M. Hall et al.: Temperature trends at 90 km over Svalbard, Norway (78 N, 16 E), seen in one decade of meteor radar observations. J. Geophys. Res., 117, D08104, doi: 10.1029/2011JD017028, 2012: No change of 90 km temperature trend over Svalbard around 2006. Meteor radar temperatures.**
   **I.A. Mokhov, A.I. Semenov: Joint analysis of the long-term behavior of temperature in the mesopause and on the Earth's surface during the period of about 60 years. 6th NDMC Meeting, Grainau, 2014 (your co-author D. Offermann attended): 1960- 2012 data from Zvenigorod near Moscow. OH mesopause temperatures. Drop in the second half of the 1970s, than negative trend till the mid-1990s and essentially no trend afterwards. No change reported around 2006.**
   **C.-Y. She, D.A. Krueger, T. Yuan: Long-term midlatitude mesopause region tem- perature trends deduced from quarter century (1990-2014) Na lidar observations. Ann. Geophysicae, 33, 363-369, doi: 10.5194/angeocom-33-363-2015, 2015. Fort Collins/Logan, USA. 85-90 km temperature trend 1990-2014 is negative, after the mid- 1990s weaker than before. No change of trend reported around 2006.**

   We added comparisons with the mentioned publications (except for Mokhov et al., since it is not published) to the corresponding section dealing with linear trends and trend breaks (Sect. 5.2).

**2 Comments Referee 2**

The paper seeks to explore an alternative explanation for the change in slope of the OH temperature time series previously reported in earlier publications using a shorter time series. While the results are promising, I believe there is an overall shortcoming in the analysis technique. While this may not substantially change the overall results, it does put into question their statistical significance.

1. When fitting data using an orthogonal basis set, one can fit the functions simultaneously or sequentially, where the first function is subtracted from the data and the next fit to the residuals. However, in this paper there is a combination orthogonal functions (sinusoids in the case of the periodogram) and non-orthogonal functions (solar cycles and trends) fitted to the data sequentially. For example in Section 4.1, the solar 10.7 cm flux and a trend first are correctly fitted simultaneously as they are non-orthogonal. However, on page 7, lines 1-13, a periodogram, which fits a set of orthogonal sinusoids, is performed on the data that has had the solar and trend subtracted. It is not technically valid to perform the LSP of the residuals since functions non-orthogonal to the resulting periodogram sine waves have been removed. In particular, on line 5, it is stated that the LSP shows a peak with a period of 20 years remains after subtraction of the solar cycle and trend. However, the subtraction of a linear trend will act as a low pass filter (Kennedy, JGR 85, p219, 1980). Thus, the "peak" at 20 years is likely an artifact of passing the original data with its red-noise through this low pass filter.

   Though the paper tries to justify this referring to Horne et al., 1986, it should be noted that Hornes's analysis removed a function orthogonal to the other periodogram components. That is not the case here. Additionally, in Horne's analysis, the estimate of variance after removal of the orthogonal function must be used to re-normalize the periodogram. That estimate of variance must be adjusted to the reduction in the number of degrees of freedom associated with the removal of these "correct" functions in much the same way that the total variance of the raw data set was estimated using N-1 to account for the removal of the mean. Here, a mean, a trend, and a solar cycle (itself a combination of several orthogonal sinusoids) have been removed, and the variance must be estimated appropriately.

We agree with the referee and made some corrections. Firstly, we calculated the variance of the raw data using N-1 degrees of freedom to account for the removal of the mean. This leads to a peak with a maximum of $(N-1)/2$ in case of a single sinusoid. So the FAP is described by

$$FAP = 1 - [1 - (\frac{2z}{N-1})^{(N-3)/2}]^{N_i},$$

since the samples taken from a gaussian distribution are processed in the same way as the original data series. Secondly, we adjusted the variance of the residuals by taking into account the number of parameters used in the fit subtracted from the data. We changed Sect. 3. and corresponding text parts.

Moreover, we removed all sequentially analysed parts and fitted all non-orthogonal functions simultaneously (see below at specific comments).

Lastly, we changed all parts where LSPs for residuals were analysed. We agree that the removal of functions non-orthogonal to the LSP sinusoids influence the LSP for the residuals. Therefore, remaining peaks are not interpreted as oscillation with a specific period that remain or even are a component of the original data series. The LSPs for the residuals are only used to visualize the reduction in variance and to show the capability of different fits to reduce the large peak at 25-30 years in the periodogram for the original data series. A fit that takes into account all long term variations of the data series should remove all signals in the long periodic range of the LSP. The corresponding text parts in Sect. 4 were changed to explain this.

2. **In section 4.2, the solar F10.7 cycle is not an orthogonal function to the two linear trends. Hence, its removal before fitting the two trends will influence the trends. All of these non-orthogonal components should be fit simultaneously.**

We fitted the two linear trends and the dependence on the solar F10.7 cm flux simultaneously. The results slightly differ from the ones before. The new analyis leads to the parameters $C_{solar} = (3.3 \pm 0.9)$ K $(100\,\mathrm{SFU})^{-1}$, $C_{trend1} = (-0.24 \pm 0.07)$ K year$^{-1}$, and $C_{trend2} = (0.64 \pm 0.33)$ K year$^{-1}$. The break point is BP $= (2008.8 \pm 1.7)$ years. Thus, all of the results agree with the former results within the combined uncertainties. We changed the corresponding section 4.2.

3. **Similarly, in section 4.2, a sinusoid is fitted to the residual after removal on non-orthogonal components. It should be fit simultaneously with the other terms as in Equation 8.**

The fit of the sinusoid is a hint to the best fitting period and amplitude of the oscillation. Thus, we kept this fit, but we fitted the oscillation together with the solar flux dependency and allowed the period to vary to derive the final result. See below point 6.

4. **Page 9, line 10, it is not mentioned whether this procedure explains more of the variance than the trend-break (assuming the trend break analysis is done correctly).**

We now calculated the $r^2$ of the trend break fit (0.74) and compared it to the other fits.

5. **Page 9, line 13. As previous comments, one has not removed orthogonal functions nor adjusted the estimate of variance to account for the removal of these functions. Thus the LSP of the residual cannot be interpreted as periodic components remaining in the data. These may well have been created by the removal of the non-orthogonal functions.**

We changed the corresponding text part (see point 1).

6. **Page 9, line 20. Again, the peak at 24 years in the periodogram could well be the result of the non-orthogonal functions being used to fit the data, and does not justify fixing the period to be fixed at 24 years in the fitting procedure. Thus, the period should also be allowed to vary in the fit in equation 9. Otherwise, there is no unambiguous evidence from the periodogram that it is exactly 24 years.**

We have done the fitting procedure again and allowed the period to vary.

$$T_0(SF, t) = C_{solar} \cdot SF + A \cdot \sin(\frac{2 \cdot \pi}{P}(t + \phi)) + b$$

The results of the new fit are $C_{solar} = (4.1 \pm 0.8)$ K $(100\,\mathrm{SFU})^{-1}$, $A = (1.95 \pm 0.43)$ K, and $P = (24.8 \pm 3.2)$ years. Thus, all of the results are almost identical to the results obtained by fixing the period to 24 years. The $r^2$ of the new fit is only slightly larger than before with a value of 0.775 (before 0.774).

7. **In Section 4.4, , the LSP of the residuals is again not valid (as stated above), and these periods should not be fixed. One could fit a series of sinusoids simultaneously with the solar cycle, where the periods are also treated as free parameters. Using a non-linear least squares fit, the periodogram of the residuals may be used to estimate the initial values of these parameters. However, it is unclear whether there are enough degrees of freedom in the 25 data points to accommodate this analysis. I would advise that the fitting be performed simultaneously rather than sequentially to avoid the problems associated with the non-orthogonal basis set. In this way, the statistical significance of the periods, and the values of the periods themselves may be assessed. If indeed the results are substantially the same, then the interpretation will hold.**

We fully agree with this comment and deleted Sect. 4.4 and also the discussion in Sect. 5.4, since a 25 data points series may not have enough degrees of freedom for a complete analysis of the MAOs together with the solar cycle and other components.

[revised manuscript text omitted]

---

## Referee Report (RR1)

I appreciate the authors taking into account my previous comments. There are minor grammatical errors that the copy editor will undoubtedly address. Now that the authors have clarified the text, there are some additional, relatively minor items that should be added prior to publication. Firstly, since the shape of the residual is being used as a measure of the need for additional fitting parameters, error bars should be placed on the residuals in Figures 6, 8 and 10 (these should be the sqrt(fitting_error^2 + temperature_error^2).

Similarly, the final conclusion that the 11-year solar cycle + a long period oscillation is the superior fit relies on the increase in the correlation coefficient, $r^2$, as one progresses from fits using the 11-year solar cycle with 1) linear trend ($r^2=0.6$); 2) two linear trends ($r^2 = 0.74$); 3) the Hale magnetic field ($r^2 = 0.71$; and 4) the long-period oscillation ($r^2 = 0.78$) . However, the error in the correlation coefficient is not given (the 95% confidence interval of the correlation coefficient is calculated in many standard statistical packages). These should be given to ensure that the increases in the correlation coefficients are statistically significant.

Thus, the intonation that the long period oscillation fits better should be justified as above. If there are no statistically significant differences in the correlation coefficients, the conclusion must be that the different fits are equivalent. However, the author's main point that a long-period oscillation fits at least as well (and perhaps significantly better) than two linear trends would still stand and represent an important contribution. Thus, I trust that the authors would correct the text accordingly, and I would not need to review the manuscript before publication.

---

## Author Response (AR2)

Reply to the reviewer comments on the paper

"Long term dynamics of OH* temperatures over Middle Europe: Trends and solar correlations"

by C.Kalicinsky et al.

We thank the referees for their helpful comments and recommendations. In the following, we discuss the issues addressed by the referees and explain our opinions and the modifications of our manuscript.

We enumerate the referee comments and repeat them in bold face. The modifications of the manuscript are displayed in the marked-up manuscript version as colored text. Deleted parts are shown in red and new text parts in blue.

**1 Comments Referee 1**

**Author responded adequately to my comments and I am satisfied by his response. Therefore I recommend to publish this paper after minor revision (no further review is necessary), even though the paper is controversial. The paper will be impulse for further discussion about origin of long-term trends, even though I personally would interpret the results of the paper in other way.**

1. **Figure 1: The vertical dashed line marking the date of Mt. Pinatubo eruption is missing in Fig. 1. With respect to the level of solar activity and to Fig. 3 I would expect somewhat lower values in years 2014 and 2015 after long gap.**

   We increased the thickness of the vertical dashed line in Fig. 1.

   In Fig. 3 there are other years where the temperatures at both stations agree (2005, 2007, 2011) and also one date where the Hohenpeissenberg temperature is lower than that at Wuppertal (2004). The fact that the two last data points for Wuppertal lie at or above the Hohenpeissenberg data points is therefore not untypical.

2. **Figure 7: Dashed black horizontal line is missing in the Figure.**

   We fixed this.

3. **Delete large sentence on page 16, line 30 – trend break for majority studied physical quantities it is natural and physical (e.g. due to change of ozone trend).**

   We deleted the corresponding text part.

4. **Wording and misprints: - Page 8, line 24: "mean the resulting" should be "mean, the resulting" – comma makes the sense of this sentence correct. - Page 13, line 10: "to large" should be "too large".**

   We fixed this.

**2 Comments Referee 2**

The improvement in residuals and correlation coefficients are used as a metric of improvement with different basis vectors used in the fitting. However, these are currently quoted without uncertainties. This is a very minor revision since these are easy to include and would not impact the authors' main conclusion that the temperature trends can be represented by a solar cycle along with either two trends or a long-period oscillation.

I appreciate the authors taking into account my previous comments. There are minor grammatical errors that the copy editor will undoubtedly address. Now that the authors have clarified the text, there are some additional, relatively minor items that should be added prior to publication.

1. **Firstly, since the shape of the residual is being used as a measure of the need for additional fitting parameters, error bars should be placed on the residuals in Figures 6, 8 and 10 (these should be the sqrt(fitting_error$^2$ + temperature_error$^2$).**

   We calculated the one sigma uncertainties $\sigma_{fit}$ of the fits using the Jacobian and the covariance matrix of the fit parameters. These uncertainties are displayed as red area around the fits. In the residual plots we decided to show the uncertainties of the fit and of the data points seperately. The fit uncertainties are shown as gray area around the zero line and the uncertainties of the data points are shown as error bars on the residual points. One has to keep in mind here, that even though the fit uncertainty shows the range of possible fits, the general shape of the fit cannot change. The shape still depends on the parameters such as solar flux, time, magnetic field etc. and so not all combination of points in the gray and reddish area are possible for a fit. Thus, the fit with one linear trend is still a relatively poor fit and cannot capture the long term variation of the data points.

   We rephrased the corresponding text parts.

2. **Similarly, the final conclusion that the 11-year solar cycle + a long period oscillation is the superior fit relies on the increase in the correlation coefficient, r$^2$, as one progresses from fits using the 11-year solar cycle with 1) linear trend (r$^2$ = 0.6); 2) two linear trends (r$^2$ = 0.74); 3) the Hale magnetic field (r$^2$ = 0.71); and 4) the long-period oscillation ($r^2$ = 0.78) . However, the error in the correlation coefficient is not given (the 95% confidence interval of the correlation coefficient is calculated in many standard statistical packages). These should be given to ensure that the increases in the correlation coefficients are statistically significant.**

   **Thus, the intonation that the long period oscillation fits better should be justified as above. If there are no statistically significant differences in the correlation coefficients, the conclusion must be that the different fits are equivalent. However, the author's main point that a**

**long-period oscillation fits at least as well (and perhaps significantly better) than two linear trends would still stand and represent an important contribution. Thus, I trust that the authors would correct the text accordingly, and I would not need to review the manuscript before publication.**

We did some simulations and used an approximative formula from literature to look at the possible range of the $r^2$ for the fits. As result there is no statistically significant difference between the trend break fit and the fit using the solar cycle and the oscillation. But the $r^2$ does not take into account the uncertainties of the annual average temperatures. It is only based on the variance of the residuals. Thus, the $r^2$ does not tell everything. So we use the LSP of the residuals and the shape of the residuals in comparison to the uncertainties of the annual average temperatures as additional measures. This shows, that using only one linear trend is not sufficient. But the additional measures does not give significant information wether the trend break description or the long-period oscillation is better.

Thus, we rephrased the corresponding parts and state, that the two descriptions lead to equivalent results.

[revised manuscript text omitted]